# Autofluorescence-Based Investigation of Spatial Distribution of Phenolic Compounds in Soybeans Using Confocal Laser Microscopy and a High-Resolution Mass Spectrometric Approach

**DOI:** 10.3390/molecules27238228

**Published:** 2022-11-25

**Authors:** Mayya P. Razgonova, Yulia N. Zinchenko, Darya K. Kozak, Victoria A. Kuznetsova, Alexander M. Zakharenko, Sezai Ercisli, Kirill S. Golokhvast

**Affiliations:** 1Far Eastern Experimental Station, N.I. Vavilov All-Russian Institute of Plant Genetic Resources, 190000 Saint-Petersburg, Russia; 2SEC Nanotechnology, Polytechnic Institute, Far Eastern Federal University, 690922 Vladivostok, Russia; 3Laboratory of Biochemistry, Blagoveshchensk State Pedagogical University, 675000 Blagoveshchensk, Russia; 4Laboratory of Pesticide Toxicology, Siberian Federal Scientific Center of Agrobiotechnology RAS, 633501 Krasnoobsk, Russia; 5Department of Horticulture, Agricultural Faculty, Ataturk University, Erzurum 25240, Turkey

**Keywords:** *Glycine* Willd, flavonols, laser microscopy, HPLC-MS/MS, tandem mass spectrometry, polyphenols

## Abstract

In this research, we present a detailed comparative analysis of the bioactive substances of soybean varieties k-11538 (Russia), k-11559 (Russia), k-569 (China), k-5367 (China), k-5373 (China), k-5586 (Sweden), and Primorskaya-86 (Russia) using an LSM 800 confocal laser microscope and an amaZon ion trap SL mass spectrometer. Laser microscopy made it possible to clarify in detail the spatial arrangement of the polyphenolic content of soybeans. Our results revealed that the phenolics of soybean are spatially located mainly in the seed coat and the outer layer of the cotyledon. High-performance liquid chromatography (HPLC) was used in combination with an amaZon SL BRUKER DALTONIKS ion trap (tandem mass spectrometry) to identify target analytes in soybean extracts. The results of initial studies revealed the presence of 63 compounds, and 45 of the target analytes were identified as polyphenolic compounds.

## 1. Introduction

*Glycine* Willd (soybean) is an economically important member of the Fabaceae family. The center of origin of the soybean is located in East Asia [1], where it has been used as food for more than 5000 years [2]. As a well-known source of cheap concentrated protein and vegetable oil, soybean occupies a place of world importance among crops. Accounting for a 53% global production share of all oilseed crops, soybean occupies a significant place in the agricultural production systems of most major countries, such as the USA, China, Brazil, Argentina, and India [3]. In recent years, soybean production in Russia has shown stable growth due to the expansion of crop acreage. In total, Russia produced more than 3 million tons of soybeans in 2016 [4].

There has been considerable interest among researchers and consumers in the potential role of soybean and soy foods in the prevention of diseases. Clinical and scientific evidence has revealed the medicinal benefits of the components of soybean against metabolic disorders and other chronic diseases (cardiovascular diseases, diabetes, obesity, cancer, osteoporosis, menopausal syndrome, anemia, etc.) [2]. As a step toward understanding the mechanisms of the influence of the food components on health, it is important to investigate chemical compositions to reveal the active components responsible for beneficial effects. It was shown that the health benefits of soybean are due to its secondary metabolites, such as isoflavones, phytosterols, lecithins, saponins, etc. [2]. In particular, Omoni et al. (2005) pointed out that isoflavones appear to work in conjunction with proteins to protect against cancer, cardiovascular disease, and osteoporosis [5].

In addition, for various crops, a relationship between the presence of phenolic compounds and the degree of plant resistance to adverse environmental conditions has been reported. Phenolic acids are important secondary plant metabolites that function as cell wall structural components, biosynthesis intermediates, and signaling and defense molecules [6]. Flavonoids, including chalcones, flavanols, flavones, flavonols, and anthocyanins, usually accumulate in the epidermal layer of plants. They are associated defense responses to ultraviolet radiation and other abiotic and biotic stresses. Thus, flavonoid distribution in the epidermal layer is an important factor for plant survival in stressful environments and is indispensable to understand the mechanisms underlying stress response and tolerance in living plant tissues and cells [7].

Polyphenolic compounds, including phenolic acids and their derivatives, tannins, and flavonoids, represent the largest group of natural plant nutrients. They determine the color of fruits and seeds and play an important role in disease resistance [8]. In soybean, the concentrations of phenolic compounds such as flavonoids and anthocyanins correlate with seed coat color [9].

One of the most important classes of phenolics is anthocyanins, which are well known for their antioxidant activity [10]. In connection with the considerable potential of anthocyanins as components of functional nutrition, knowledge about their genetic control is in demand, as they are used in breeding programs aimed at creating new varieties of cultivated plants with an increased content of these compounds that are valuable for human health. Unfortunately, as crops are cultivated, a significant portion of their biodiversity is lost, which explains the increased research interest in the study of the biodiversity of wild forms of various crops.

New progressive research methods are becoming more widespread, such as laser microscopy, a method that exploits the ability of chemicals to fluoresce when excited by a laser and can be used to solve problems of visualization. Currently, microscopic images are successfully used to visualize the location of chemicals in organs and tissues of various plants [11,12]. However, previous autofluorescence-based microscopic studies of soybean were focused on visualization of anatomical features, such as the three-dimensional (3D) internal structure of a soybean seed [13] and the leaf anatomy of *Glycine max* (L.) Merr. [14].

Although the use of various microscopy methods is common in the study of soybeans, most of these approaches focus only on optical microscopy, specific staining of proteins or polysaccharides, and analysis of the signals of specific antibodies with a fluorescence label [15,16,17].

Therefore, we investigated the polyphenolic composition of soybean, in particular anthocyanins, and showed their localization in seeds based on the autofluorescence. Such a simple method as recording autofluorescence signals is significantly underestimated and can provide a sufficiently large amount of information without complex sample preparation. Despite the insufficiency of using this method without the support of deeper analysis data, such as RAMAN spectroscopy or MALDI spectrometry, in this study, we show that the method is applicable to deeper analysis of seeds in terms of classes of compounds present and that the obtained data correlate with more complex methods. Thus, the proposed method promising for obtaining preliminary data and analyzing a large number of varietal samples. The use of this approach is time- resource-, and reagent-saving and can help to increase the level of research in laboratories that do not have more complex equipment.

## 2. Materials and Methods

### 2.1. Materials

As an object of research, we used the following soybean varieties cultivated at the N.I. Vavilov All-Russian Institute of Plant Genetic Resources: k-11538 (*G. soja,* cultivated form OLMIK-76, Russia), k-11559 (*G. soja*, wild, Russia), k-569 (*G. gracilis*, China), k-5367 (*G. gracilis*, E-Shen-Dow, China), k-5373 (*G. gracilis*, Harbin semiwild, China), k-5586 (*G. max,* 856-3-3, Sweden), and Primorskaya-86 (*G. max*, Russia).

Seeds from the VIR collection were selected, and the maximum diversity in appearance was taken into account. Seeds were obtained from the research fields of the N. I. Vavilov All-Russian Institute of Plant Genetic Resources (VIR) according to the developed VIR Guidelines. Because the purpose of this study was to investigate the diversity of polyphenolic compounds of soybean, the 5 most colored varieties and two control light-colored varieties were selected from the VIR collection (Figure 1).

### 2.2. Chemicals and Reagents

HPLC-grade acetonitrile was purchased from Fisher Scientific (Southborough, UK), and MS-grade formic acid was obtained from Sigma-Aldrich (Steinheim, Germany). Ultrapure water was prepared using a SIEMENS ULTRA clear (SIEMENS water technologies, Munich, Germany), and all other chemicals were of analytical grade. The results were obtained using the equipment of the Center for Collective Use of Scientific Equipment of the Tambov State University named after G.R. Derzhavin.

### 2.3. Fractional Maceration

A fractional maceration technique was applied to obtain highly concentrated extracts [18]. From 500 g of the sample, 1 g of soy seeds was randomly selected for maceration. The total amount of the extractant (reagent-grade methyl alcohol) was divided into three parts and consistently infused with the grains with the first, second, and third parts with a solid–solvent ratio of 1:20. The infusion of each part of the extractant lasted 7 days at room temperature.

After maceration, the samples were centrifuged to precipitate sediment at an acceleration of 5000× *g* and a temperature of 4 °C for 20 min; then, a 3 mL aliquot of the sample was filtered on syringe filters with a pore size of 0.45 μm, and the first 2 mL of filtrate was discarded for non-specific sorption on the membrane, and only the last milliliter was used for analysis. The filtered milliliter of the sample was diluted with 1 mL of deionized water.

### 2.4. Optical Microscopy 

Dry, untreated soybean seeds were used for confocal laser scanning microscopy. The transverse dissection of seeds was performed with an MS-2 sled microtome (Tochmedpribor, Kharkiv, Ukraine). The autofluorescence parameters were determined using an inverted confocal laser scanning microscope in lambda mode (LSM 800, Carl Zeiss Microscopy GmbHAG, Jena, Germany). We carried out a lambda experiment with excitation lasers at 405, 488, 561, and 740 nm and registered emissions in the range of 400 to 700 nm with a step of 5 nm. The maxima of fluorescence were registered with the following parameters: excitation by a violet laser (405 nm) with emission in the range of 400–475 nm (blue); excitation by a blue laser (488 nm) with the emission in the range of 500–545 nm (green) and 620–700 nm (red). Images were obtained using 63× magnification and ZEN 2.1 software (Carl Zeiss Microscopy GmbH, Jena, Germany).

### 2.5. Liquid Chromatography

HPLC was performed using an LC-20 Prominence HPLC (Shimadzu, Kyoto, Japan) equipped with a UV sensor and a C_18_ silica reverse-phase column (4.6 × 150 mm, particle size: 2.7 µm) for separation of multicomponent mixtures. A gradient elution program with two mobile phases (A, deionized water; B, acetonitrile with formic acid 0.1% *v*/*v*) was performed as follows: 0–2 min, 0% B; 2–50 min, 0–100% B; control washing 50–60 min, 100% B. The entire HPLC analysis was performed with an SPD-20A UV-vis detector (Shimadzu, Japan) at wavelengths of 230 nm and 330 nm; the temperature was 50 °C, and the total flow rate was 0.25 mL min^−1^. The injection volume was 10 µL. Additionally, liquid chromatography was combined with a mass spectrometric ion trap to identify compounds.

### 2.6. Mass Spectrometry

MS analysis was performed on an amaZon SL ion trap (BRUKER DALTONIKS, Bremen, Germany) equipped with an ESI source in negative and positive ion mode. The optimized parameters were obtained as follows: ionization source temperature, 70 °C; gas flow, 9/min; nebulizer gas (atomizer), 7.3 psi; capillary voltage, 4500 V; end-plate bend voltage, 1500 V; fragmentary voltage, 280 V; collision energy, 60 eV. An ion trap was used in the scan range of *m*/*z* 100–1.700 for MS and MS/MS. All experiments were repeated three times. A four-stage ion separation mode (MS/MS mode) was implemented.

## 3. Results and Discussion

### 3.1. Optical Microscopy of Soybean Components

The observation of autofluorescence makes it possible to draw conclusions about the presence and localization of fluorescent substances in plant tissues. An increased level of fluorescence signal in individual areas reflects the main accumulation sites of certain classes of compounds. Figure 2, Figure 3, Figure 4, Figure 5, Figure 6, Figure 7 and Figure 8 show transverse sections of soybean seeds under a confocal laser microscope. Microscopic examination showed the presence of fluorescent substances in the soybean seeds.

We observed three main autofluorescence maxima: in the blue (400–475 nm), green (500–545 nm), and red (620–700 nm) regions of the spectrum. According to the literature data, the blue fluorescence in plants is mainly due to the presence of phenolic hydroxycinnamic acids [19]. The main fluorescent component is ferulic acid, but other hydroxycinnamic (e.g., p-coumaric and caffeic) acids can also contribute to fluorescence [20]. Moreover, lignin is a well-known source of blue fluorescence in plants. It has a wide emission range, owing to the presence of multiple fluorophore types within the molecule and can be observed when excited by UV and visible light [21]. Previous studies have shown that the lignin content of legume seed coat is low [22,23] and that the cotyledons are poorly lignified [24]. Therefore, we concluded that most of the blue fluorescence in soybean seeds comes from hydroxycinnamic acids.

The blue-light-induced green autofluorescence in the range of 500–545 nm can be explained by the presence of flavins and flavonols (myricetin, quercetin, and kaempferol) and their derivatives [7,25,26]. The emission in the red spectrum mainly occurs due to the presence of anthocyanins and anthocyanidins [27,28].

We studied the seeds of three different soybean species (*G. soja* in cultivated and wild forms, as well as *G. gracilis* and *G. max*) and found that the spatial distribution of fluorescent substances has features that correlate with the color of the seeds.

In general, our study showed the maximum of blue fluorescence, which reflects the content of hydroxycinnamic acids, in the outer cotyledon layer. A weaker signal was observed in the rest of the cotyledon parenchyma cells. In the seed coat of the dark-colored seeds, the signal was almost absent. On the contrary, the light-colored seeds (yellow) showed a solid blue signal (Figure 7a and Figure 8a). Similar results were obtained in the other studies on the chemical composition of legume seeds. It was reported that coumaric and ferulic acids are dominant phenolic acids in the white seed coat of pea, as compared with colored seed coats [29].

Green fluorescence is most pronounced in the outer layer of the cotyledon. The signal is also present in the seed coat but it is usually weaker than that in the outer layer. The brightest green fluorescence of the palisade layer of the seed coat is observed in yellow seeds. This fluorescence is the most expressed among all investigated varieties and comparable to that of the outer cotyledon layer (Figure 7b and Figure 8b).

The level of the red fluorescence signal correlates well with the color of the seeds. Microscopic examination showed that the palisade layer of black-seeded varieties has the brightest red fluorescence, whereas yellow-seeded varieties have the weakest red fluorescence. The brown-seeded variety demonstrated red fluorescence in the form of scattered inclusions (Figure 5c). It was previously reported that the black color of the seed coat in legumes is the result of a large amount of anthocyanins [30]. This confirms that bright red fluorescence is caused by such chemicals.

Our result show that various phenolic substances are responsible for autofluorescence in soybean. The total fluorescence signal is maximal in the seed coat of all varieties. Our results are consistent with numerous publications indicating that the total concentration of phenolic compounds is always much higher in the seed coat than in the cotyledons of legumes [31,32]. The accumulation of phenolics mainly in the outer layers of the seed may be associated with their protective function during seed development, as well as their protective function against detrimental agents in the environment [33].

### 3.2. Tandem Mass Spectrometric Analysis

The most-consumed extracts of soybeans were analyzed by HPLC-MS/MS ion trap to better interpret the diversity of available phytochemicals. All of the examined extracts have a rich bioactive composition. Each compound was structurally identified on the basis of their accurate mass and MS/MS fragmentation by HPLC-ESI ion trap MS/MS. Sixty-three biologically active compound were successfully identified and characterized by comparing fragmentation patterns and retention times. Other compounds were identified by comparing their MS/MS data with available literature data. All identified compounds, along with molecular formulae, calculated and observed *m*/*z*, MS/MS data, and their comparative profile for soybeans (seven varieties), are summarized in Table 1.

In the present study, 45 polyphenolic compounds were identified and characterized, including 17 flavones, 10 flavonols, 3 flavan-3-ols, 1 flavanone, 3 anthocyanidins, 3 condensed tannins, 5 phenolic acids, 1 lignan, 1 stilbene, and 1 hydroxycoumarin. Additionally, 18 compounds of other classes were identified in soybeans, with some identified for the first time, for example, steroidal alkaloids Alpha-chaconine and solanidadiene solatriose. Table 2 lists the identified polyphenolic compounds in seven varieties of soybeans. In our research, the richest polyphenolic content was observed in the Chinese variety k-5373 (Harbin semiwild). In this variety, 30 polyphenolic compounds were identified during primary studies. The Russian variety k-11538 (OLMIK-76) is in second place in terms of the richness of compounds, with 23 compounds identified.

Figure 9 and Figure 10 show examples of the decoding spectra (collision-induced dissociation (CID) spectrum) of the ion chromatogram obtained using tandem mass spectrometry. The mass spectrum in positive ion mode of Cyanidin 3-*O*-glucoside from extracts of soyabean k-5373 (China, Harbin semi-wild) is shown in Figure 9. The [M + H]^+^ ion produced one fragment ion at *m/z* 287. The fragment ion with *m/z* 287 yielded two daughter ions at *m/z* 213 and *m/z* 137. This compound was identified in the bibliography as cyanidin 3-*O*-glucoside in extracts from *Clidemia rubra* [82], *Triticum* [40,101], *acerola* [60], rice [65], Disterigma [43], *Vigna sinensis* [102], *Vitis labrusca* [103], and rapeseed petals [71].

The mass spectrum in positive ion mode of proanthocyanidin B1 from extracts from extracts of soyabean k-5373 (China, Harbin semi-wild) is shown in Figure 10. The [M + H]^+^ ion produced five fragment ions at *m/z* 409, *m/z* 343, *m/z* 291, *m/z* 247, and *m/z* 205. The fragment ion with *m/z* 409 yielded four daughter ions at *m/z* 287, *m/z* 259, *m/z* 203, and *m/z* 163. The fragment ion with *m/z* 287 yielded two daughter ions at *m/z* 245 and *m/z* 203. To the best of our knowledge, proanthocyanidin B1 has been reported in millet grains [41], pear [108], *Vaccinium macrocarpon* [73], Andean blueberry [43], strawberry [74], *Vigna inguiculata* [49], *Senna singueana* [109], *Camellia kucha* [37], grape juice [107], vinery products [52], etc.

## 4. Conclusions

The results of a preliminary study showed the presence of 63 compounds corresponding to the Glycine Willd genus (soybean), some of which were identified for the first time in Glycine. The extracts of soybean k-5373 (China, Harbin semi-wild) contain the most polyphenolic complexes, which are biologically active compounds. Laser microscopy made it possible to clarify in detail the spatial arrangement of the polyphenolic content of soybeans. Results showed that phenolics of soybean are spatially located mainly in the seed coat and the outer layer of the cotyledon. Anthocyanins are especially abundant in the palisade layer of dark-colored varieties. The seed coat of yellow-seeded varieties contains more phenolic acids and flavonols than the seed coat of dark-seeded varieties. This information can be useful for rapid evaluation of varieties for selection and breeding with respect to those compounds.

## Figures and Tables

**Figure 1 molecules-27-08228-f001:**
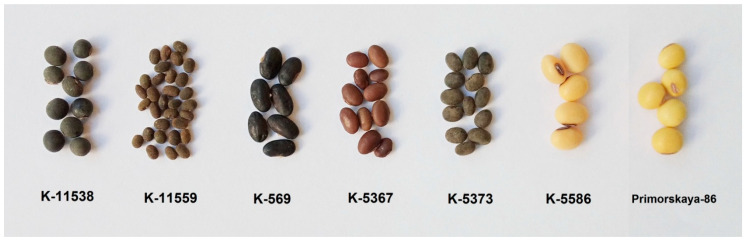
Soybean varieties k-11538 (Russia), k-11559 (Russia), k-569 (China), k-5367 (China), k-5373 (China), k-5586 (Sweden), and Primorskaya-86 (Russia).

**Figure 2 molecules-27-08228-f002:**
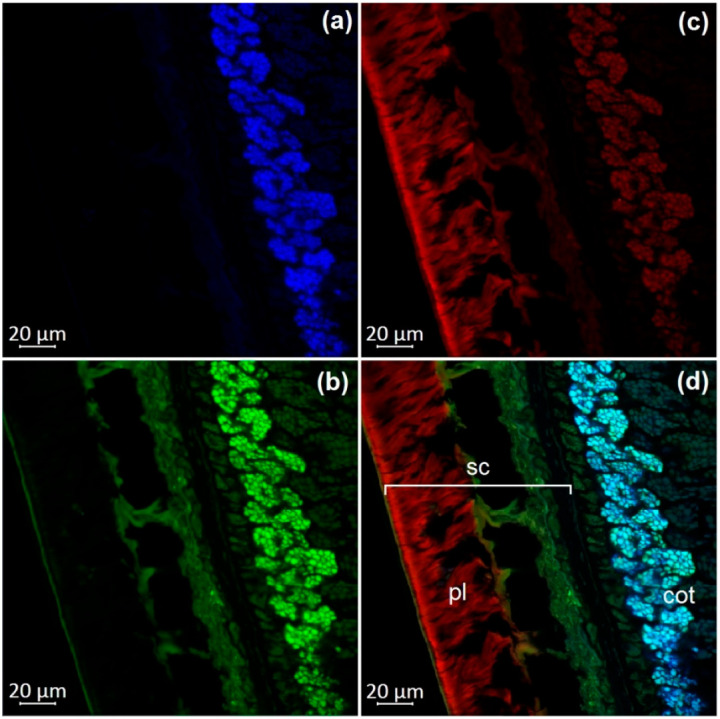
A transverse section of a soybean seed (variety k-11538): (**a**) excitation at 405 nm with emission in the range of 400–475 nm (blue); (**b**) excitation at 488 nm with emission in the range of 500–545 nm (green); (**c**) excitation at 488 nm with emission in the range of 620–700 nm (red); (**d**) merged; cot, cotyledon; pl, palisade layer; sc, seed coat.

**Figure 3 molecules-27-08228-f003:**
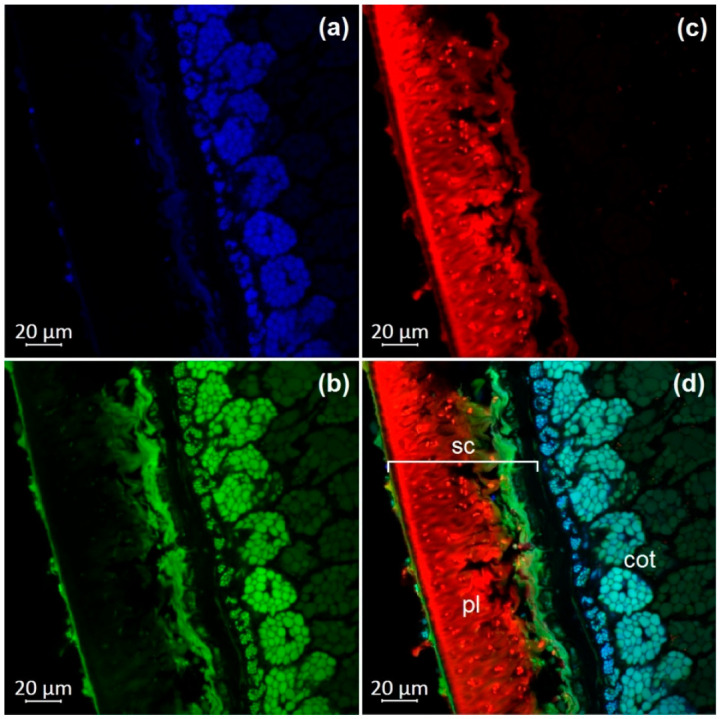
A transverse section of a soybean seed (variety k-11559): (**a**) excitation at 405 nm with emission in the range of 400–475 nm (blue); (**b**) excitation at 488 nm with emission in the range of 500–545 nm (green); (**c**) excitation at 488 nm with emission in the range of 620–700 nm (red); (**d**) merged; cot, cotyledon; pl, palisade layer; sc, seed coat.

**Figure 4 molecules-27-08228-f004:**
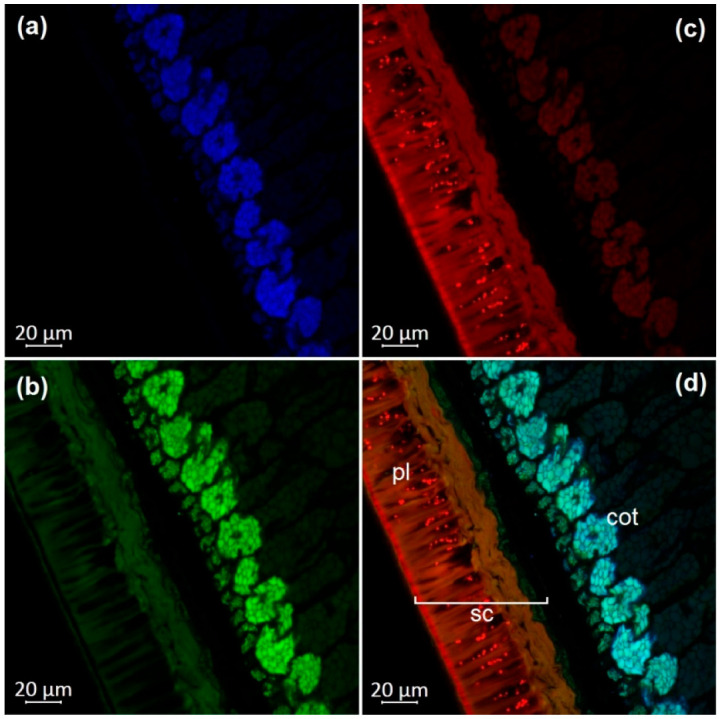
A transverse section of a soybean seed (variety k-569): (**a**) excitation at 405 nm with emission in the range of 400–475 nm (blue); (**b**) excitation at 488 nm with emission in the range of 500–545 nm (green); (**c**) excitation at 488 nm with emission in the range of 620–700 nm (red); (**d**) merged; cot, cotyledon; pl, palisade layer; sc, seed coat.

**Figure 5 molecules-27-08228-f005:**
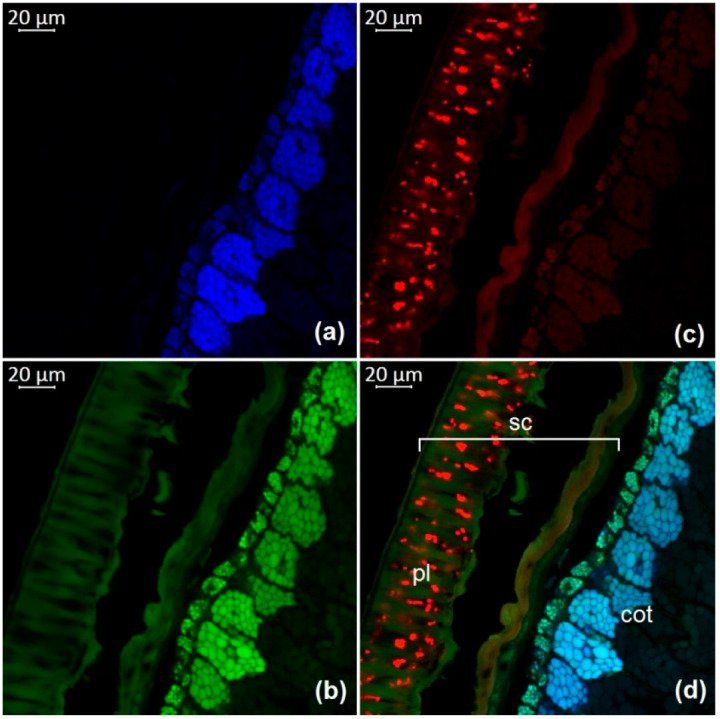
A transverse section of a soybean seed (variety k-5367): (**a**) excitation at 405 nm with emission in the range of 400–475 nm (blue); (**b**) excitation at 488 nm with emission in the range of 500–545 nm (green); (**c**) excitation at 488 nm with emission in the range of 620–700 nm (red); (**d**) merged; cot, cotyledon; pl, palisade layer; sc, seed coat.

**Figure 6 molecules-27-08228-f006:**
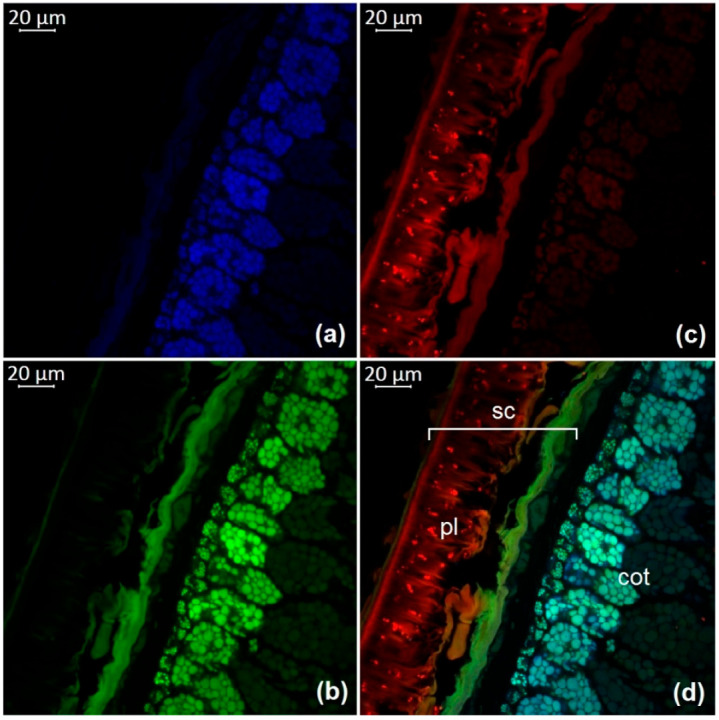
A transverse section of a soybean seed (variety k-5373): (**a**) excitation at 405 nm with emission in the range of 400–475 nm (blue); (**b**) excitation at 488 nm with emission in the range of 500–545 nm (green); (**c**) excitation at 488 nm with emission in the range of 620–700 nm (red); (**d**) merged; cot, cotyledon; pl, palisade layer; sc, seed coat.

**Figure 7 molecules-27-08228-f007:**
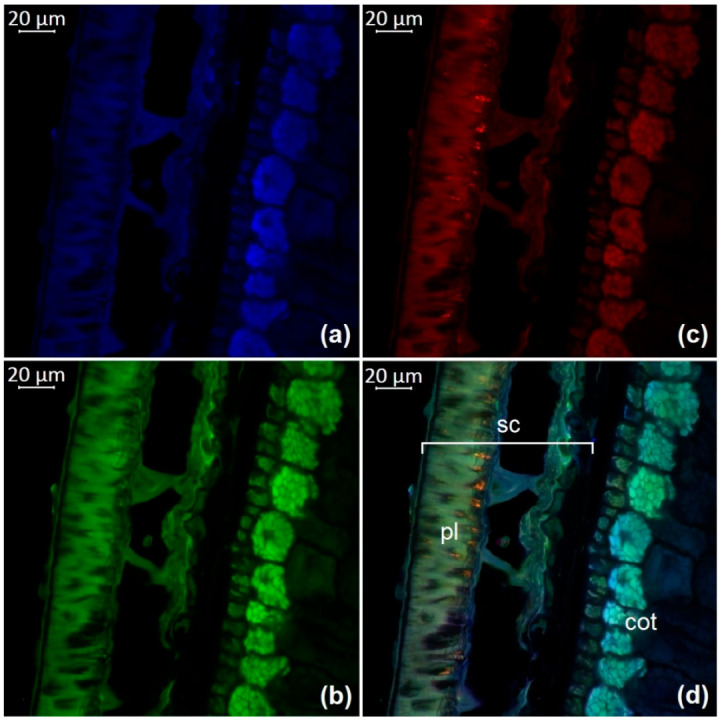
A transverse section of a soybean seed (variety k-5586): (**a**) excitation at 405 nm with emission in the range of 400–475 nm (blue); (**b**) excitation at 488 nm with emission in the range of 500–545 nm (green); (**c**) excitation at 488 nm with emission in the range of 620–700 nm (red); (**d**) merged; cot, cotyledon; pl, palisade layer; sc, seed coat.

**Figure 8 molecules-27-08228-f008:**
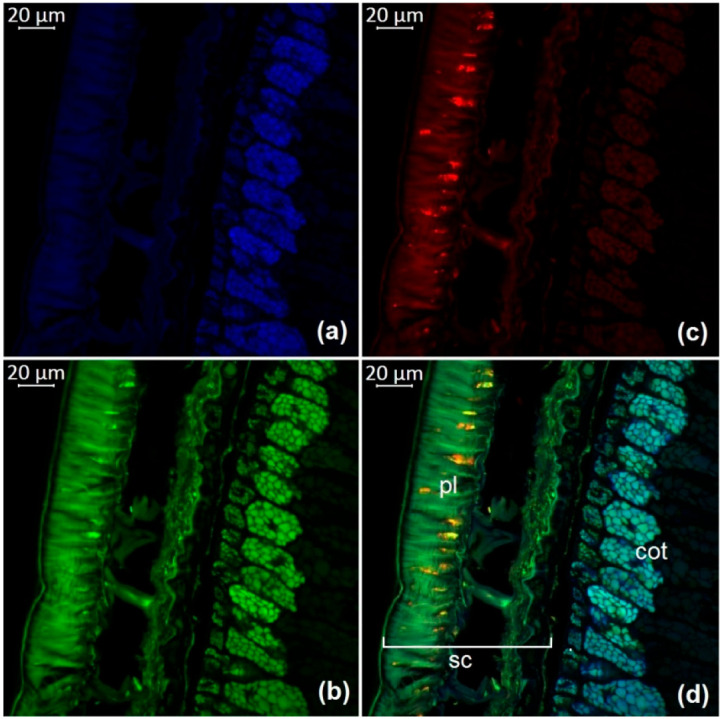
A transverse section of a soybean seed (variety Primorskaya-86): (**a**) excitation at 405 nm with emission in the range of 400–475 nm (blue); (**b**) excitation at 488 nm with emission in the range of 500–545 nm (green); (**c**) excitation at 488 nm with emission in the range of 620–700 nm (red); (**d**) merged; cot, cotyledon; pl, palisade layer; sc, seed coat.

**Figure 9 molecules-27-08228-f009:**
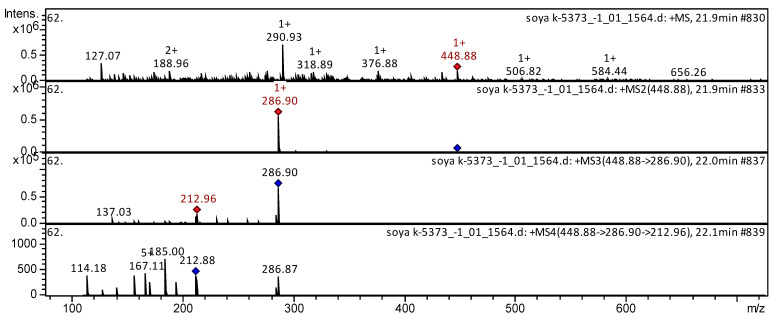
Mass spectrum of cyanidin 3-*O*-glucoside from extracts of soyabean k-5373 (China, Harbin semi-wild), *m/z* 448.88.

**Figure 10 molecules-27-08228-f010:**
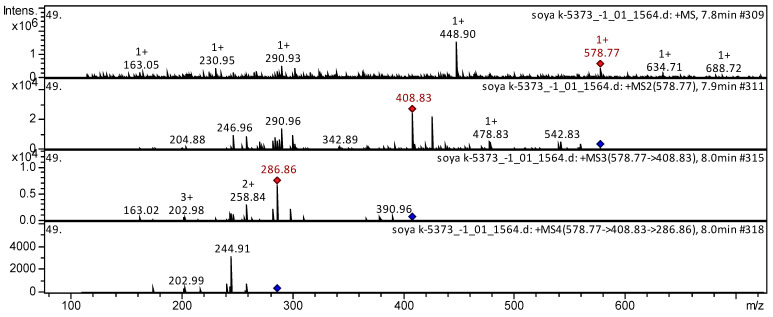
Mass spectrum of proanthocyanidin B1 from extracts of soyabean k-5373 (China, Harbin semi-wild), *m/z* 578.77.

**Table 1 molecules-27-08228-t001:** Compounds identified from the extracts of seven soybean varieties in positive and negative ionization modes by HPLC ion trap MS/MS: k-11538 (Russia), k-11559 (Russia), k-569 (China), k-5367 (China), k-5373 (China), k-5586 (Sweden), and Primorskaya-86 (Russia).

№	Class of Compound	Identified Compound	Formula	Mass	Molecular Ion [M − H]^−^	Molecular Ion [M + H]^+^	2 Fragmentation MS/MS	3 Fragmentation MS/MS	4 Fragmentation MS/MS	References
**1**	**Amino acid**	L-Leucine [(S)-2-Amino-Methylpentanoic acid]	**C_6_H_13_NO_2_**	**131.1729**		**132**	**114**			Potato leaves [34]; *Vigna unguiculata* [35]; *Lonicera japonica* [36]; *Camellia kucha* [37]
**2**	**Benzaldehyde**	Vanillin	**C_8_H_8_O_3_**	**152.15**		**153**	**151**	**136**		Potato [38,39]; *Triticum* [40]; *millet grains* [41]
**3**	**Trans-cinnamic acid**	Ferulic acid	**C_10_H_10_O_4_**	**194.184**		**195**	**177; 141**	**126**		*Lonicera japonica* [36]; Potato [38,39]; *Zostera marina* [42]; Andean blueberry [43]; Tomato [44]; *Codonopsis Radix* [45]; *Bougainvillea* [46]
**4**	**Amino acid**	**L-Tryptophan** [Tryptophan; (S)-Tryptophan]	**C_11_H_12_N_2_O_2_**	**204.2252**		**205**	**188**	**144**	**118**	*Vigna unguiculata* [35]; *Camellia kucha* [37]; *Perilla frutescens* [47]; *Passiflora incarnata* [48]; *Vigna inguiculata* [49];
**5**	**Stilbene**	**Resveratrol** [trans-Resveratrol; 3,4′,5-Trihydroxystilbene; Stilbentriol]	**C_14_H_12_O_3_**	**228.2433**		**229**	**210**	**141; 169**	**123**	*Embelia* [50]; *Red wines* [51]; vinery products [52]; *A. cordifolia*; *F. glaucescens*; *F. herrerae* [53]; *Radix polygoni multiflori* [54]
**6**	**Isoflavone**	**Daidzein** [4′,7 -Dihydroxyisoflavone; Daidzeol]	**C_15_H_10_O_4_**	**254.2375**		**255**	**227; 199; 137**	**181**		*Hedyotis diffusa* [55]; Isoflavones [56]
**7**	**Ribonucleoside composite of adenine (purine)**	**Adenosine**	**C_10_H_13_N_5_O_4_**	**267.2413**		**268**	**136**			*Lonicera japonica* [36]; Huolisu Oral Liquid [57]
**8**	**7-hydroxyisoflavone**	**Formononetin** [Biochanin B; Formononetol]	**C_16_H_12_O_4_**	**268.2641**		**269**	**254; 159; 118**	**237; 181; 118**	**237; 181**	*Astragali Radix* [45]; Isoflavones [56]; Huolisu Oral Liquid [57];
**9**	**Flavone**	**Apigenin** [5,7-Dixydroxy-2-(40Hydroxyphenyl)-4H-Chromen-4-One]	**C_15_H_10_O_5_**	**270.2369**		**271**	**153; 215**	**111**		*Lonicera japonica* [36]; millet grains [41]; Andean blueberry [43]; *Hedyotis diffusa* [55]; Mexican lupine species [58]; *Wissadula periplocifolia* [59]
**10**	**Anthocyanin**	**Pelargonidin [Pelargonidol chloride]**	**C_15_H_11_O_5+_**	**271.2493**		**271**	**215; 197; 153**	**197; 169; 141**	**169**	*acerola* [60]
**11**	**Flavan-3-ol**	**Epiafzelechin** [(epi)Afzelechin]	**C_15_H_14_O_5_**	**274.2687**		**275**	**247; 193; 147**	**193; 175**		*A. cordifolia*; *F. glaucescens*; *F. herrerae* [53]; *Cassia granidis* [61]; *Cassia abbreviata* [62]
**12**	**Omega-3 fatty acid**	**Stearidonic acid** [6,9,12,15-Octadecatetraenoic acid; Moroctic acid]	**C_18_H_28_O_2_**	**276.4137**		**277**	**217**	**190**		*G. linguiforme* [53]; *Salviae Miltiorrhizae* [63]; *Rhus coriaria* [64]
**13**	**Sceletium alkaloid**	**4′-** *O* **-desmethyl mesembranol**	**C_16_H_23_NO_3_**	**277.3587**	**276**		**234**	**218**	**218**	*A. cordifolia* [53]
**14**	**Omega-3 fatty acid**	**Linolenic acid** (Alpha-Linolenic acid; Linolenate)	**C_18_H_30_O_2_**	**278.4296**						*Salviae* [63]; rice [65]; *Pinus sylvestris* [66]
**15**	**Octadec-9-enoic acid**	**Oleic acid** (Cis-9-Octadecenoic acid; Cis-Oleic acid)	**C_18_H_34_O_2_**	**282.4614**		**283**	**209; 153**			*Zostera marina* [42]; *Sanguisorba officinalis* [67]; *Pinus sylvestris* [66]
**16**	**Flavone**	**Acacetin** [Linarigenin; Buddleoflavonol]	**C_16_H_12_O_5_**	**284.2635**		**285**	**270; 224**	**241**		Mexican lupine species [58]; *Wissadula periplocifolia* [59]; *Mentha* [68,69]; *Dracocephalum palmatum* [70]
**17**	**Flavone**	**6,7-Dihydroxy-4′-methoxyisoflavone**	**C_16_H_12_O_5_**	**284.2635**		**285**	**270; 229; 145**	**242; 152**		*Mentha* [68]
**18**	**Flavonol**	**Kaempferol** [3,5,7-Trihydroxy-2-(4-hydro- xyphenyl)-4H-chromen-4-one]	**C_15_H_10_O_6_**	**286.2363**	**285**		**257; 184; 117**	**117**		Potato leaves [34]; *Lonicera japonica* [36]; Potato [38]; Andean blueberry [43]; *Rhus coriaria* [64]; Rapeseed petals [71]
**19**	**Flavan-3-ol**	**Catechin**	**C_15_H_14_O_6_**	**290.2681**		**291**	**243; 189**	**215; 197**		Potato [39]; *Triticum* [40]; millet grains [41]; Eucalyptus [72]; *Vaccinium macrocarpon* [73]
**20**	**Flavan-3-ol**	**(epi)catechin**	**C_15_H_14_O_6_**	**290.2681**		**291**	**273; 117**	**255; 145**		millet grains [41]; *C. edulis* [53]; *Radix polygoni multiflori* [54]; *Camellia kucha* [37]
**21**	**Flavone**	**Chrysoeriol** [Chryseriol]	**C_16_H_12_O_6_**	**300.2629**		**301**	**299; 253; 152**	**226**		*Dracocephalum palmatum* [70]; *Rhus coriaria* [64]; Rice [65]; Mentha [68]; Mexican lupine species [58]
**22**	**Hydroxybenzoic acid**	**Ellagic acid** [Benzoaric acid; Elagostasine; Lagistase; Eleagic acid]	**C_14_H_6_O_8_**	**302.1926**		**303**	**275; 202**	**157**	**139**	*Rhus coriaria* [64]; strawberry [74]; *Rubus occidentalis* [75]; vinery products [52]; *Chamaecrista nictitans* [76]; *Punica granatum* [77]
**23**	**Flavonol**	**Quercetin**	**C_15_H_10_O_7_**	**302.2357**		**303**	**244; 202; 184**	**175; 156**	**129**	Potato leaves [34]; *Triticum* [40]; Tomato [44]; millet grains [41]; Red wines [51]; vinery products [52]; *Rhus coriaria* [64]; Eucalyptus [72]; *Vaccinium macrocarpon* [73]
**24**	**Flavanone**	**Hesperitin** [Hesperetin]	**C_16_H_14_O_6_**	**302.2788**		**303**	**202; 257; 185**	**156**		Andean blueberry [43]; [78]; Red wines [51]; Mentha [79]
**25**	**Diterpenoid**	**Tanshinone IIB** [(S)-6-(Hydroxymethyl)-1,6-Dimethyl-6,7,8,9-Tetrahydrophenanthro [1,2-B]Furan-10,11-Dione]	**C_19_H_18_O_4_**	**310.3438**		**311**	**292; 189; 135**	**217; 135**		*Salviae miltiorrhiza* [63]
**26**	**Flavone**	**5,7-Dimethoxyluteolin**	**C_17_H_14_O_6_**	**314.2895**	**313**		**212; 185; 113**	**113**		*Syzygium aromaticum* [80]
**27**	**Omega-hydroxy-long-chain fatty acid**	**19-Hydroxynonadecanoic acid**	**C_19_H_38_O_3_**	**314.5032**		**315**	**287; 241; 187**	**241; 187**	**169; 124**	*A. cordifolia* [53]
**28**	**Flavonol**	**Rhamnetin I** [beta-Rhamnocitrin; Quercetin 7-Methyl ether]	**C_16_H_12_O_7_**	**316.2623**		**317**	**299; 243; 189;165; 123**	**147; 123**		*Rhus coriaria* L. (*Sumac*) [64]; *Mangifera indica* [81]
**29**	**Flavonol**	**Isorhamnetin** [Isorhamnetol; Quercetin 3′-Methyl ether; 3-Methylquercetin]	**C_16_H_12_O_7_**	**316.2623**		**317**	**288; 243; 189**	**260; 242; 187**		Andean blueberry [43]; *Eucalyptus* [72]; *Astragali Radix* [45]; *Embelia* [50]; *Rapeseed petals* [71]; *Syzygium aromaticum* [80]
**30**	**Flavonol**	**Myricetin**	**C_15_H_10_O_8_**	**318.2351**		**319**	**271; 217**	**243; 189; 171**	**171**	millet grains [41]; Red wines [51]; Andean blueberry [43]; *Sanguisorba officinalis* [67]; *F. glaucescens* [53]; *Clidemia rubra* [82]
**31**	**Hydroxycoumarin**	**Umbelliferone hexoside**	**C_15_H_16_O_8_**	**324.2827**		**325**	**306; 289;225; 163**	**145**		*G. linguiforme* [53]
**32**	**Long-Chain Polyunsaturated Fatty Acid**	**Docosahexaenoic acid** [Doconexent; Cervonic acid]	**C_22_H_32_O_2_**	**328.4883**		**329**	**327; 281; 181; 115**	**199**		Marine extracts [83]
**33**	**Trihydroxyflavone**	**Jaceosidin** [5,7,4′-trihydroxy-6′,5′-dimetoxyflavone]	**C_17_H_14_O_7_**	**330.2889**		**331**	**329; 285; 231; 191; 163**	**328; 286; 216**		Mentha [68,84]
**34**	**Trihydroxyflavone**	**5,7-Dimethoxy-3,3′,4′-trihydroxyflavone**	**C_17_H_14_O_7_**	**330.2889**		**331**	**303; 185**	**157**		*Oxalis corniculata* [85]
**35**	**Flavonol**	**Myricetin 5-Methyl ether** [5-O-Methylmyricetin]	**C_16_H_12_O_8_**	**332.2617**		**333**	**287; 241; 205; 177**	**177; 149**	**149; 123**	*Vitis amurensis* [86]; *Rhodiola rosea* [87]
**36**	**Alpha, omega-dicarboxylic acid**	**Eicosatetraenedioic acid**	**C_20_H_30_O_4_**	**334.4498**		**335**	**307; 289; 233**	**277; 246; 207**		*G. linguiforme* [53]
**37**	**Flavone**	**Syringetin**	**C_17_H_14_O_8_**	**346.2883**		**347**	**317; 290; 219; 169**	**289; 272; 219**	**261; 173**	*C. edulis* [53]
**38**	**Lignan**	**Matairesinol** [(−)-Matairesinol; Artigenin Congener]	**C_20_H_22_O_6_**	**358.3851**		**359**	**325; 289; 258; 198**	**143**	**127**	*Punica granatum* [88]; Lignans [89]
**39**	**Flavone**	**5,6-Dihydroxy-7,8,3′,4′-** **tetramethoxyflavone**	**C_19_H_18_O_8_**	**374.3414**		**375**	**346; 219; 173**	**319; 273; 219; 173**	**273; 219; 173**	*Mentha* [68]
**40**	**Hydroxycinnamic acid**	**Caffeic acid derivative**	**C_16_H_18_O_9_Na**	**377.2985**	**377**		**341; 215**	**179**		Bougainvillea [46]; *Embelia* [50]
**41**	**Sterol**	**Campesterol** [Dihydrobrassicasterol]	**C_28_H_48_O**	**400.6801**		**401**	**381; 304; 225; 171**	**363; 345; 279; 225; 169**	**345; 261; 202**	*A. cordifolia*; *C. edulis* [53]
**42**	**Sterol**	**Stigmasterol** [Stigmasterin; Beta-Stigmasterol]	**C_29_H_48_O**	**412.6908**		**413**	**301; 279; 189**	**171**		*Hedyotis diffusa* [55]; *A. cordifolia*; *F. pottsii* [53]; Olive leaves [90]; *Salvia* [91]
**43**	**Sterol**	**Beta-Sitostenone** [Stigmast-4-En-3-One; Sitostenone]	**C_29_H_48_O**	**412.6908**		**413**	**395; 345; 301; 171**	**189; 171**		*F. herrerae* [53]; *Cryptomeria japonica bark* [92]; *Terminalia laxiflora* [93]
**44**	**Hydroxybenzoic acid**	**Salvianolic acid D**	**C_20_H_18_O_10_**	**418.3509**		**419**	**373; 293; 212; 127**	**329; 271; 192; 127**		*Mentha* [69,94]; *Salvia multiorrizae* [95]
**45**	**Iridoid monoterpenoid**	**Dihydroisovaltrate**	**C_22_H_32_O_8_**	**424.4847**		**425**	**365; 327; 281; 207**	**309; 253**	**235**	*Rhus coriaria* [64]
**46**	**Flavone**	**Apigenin-7-***O***-glucoside** [Apigetrin; Cosmosiin]	**C_21_H_20_O_10_**	**432.3775**		**433**	**271**	**153; 214**		Tomato [44]; *Grataegi fructus* [45]; *Mexican lupine species* [58]; *Dracocephalum palmatum* [70]; *Mentha* [84]; *Malva sylvestris* [96]
**47**	**Hydroxybenzoic acid**	**Ellagic acid pentoside** [Ellagic acid 4-*O*-xylopyranoside]	**C_19_H_14_O_12_**	**434.3073**	**433**		**257**	**227; 157**	**199; 127**	*Strawberry* [74]; *Chamaecrista nictitans* [76]; *Punica granatum* [77]; *Rubus ulmifolius* [97]
**48**	**Flavonol**	**Dihydrokaempferol-3-** *O* **-rhamnoside**	**C_21_H_22_O_10_**	**434.3934**	**433**		**259**	**258; 229**	**199**	*Vitis vinifera* [98]
**49**	**Dihydroflavonol**	**Aromadendrin 7-O-rhamnoside**	**C_21_H_22_O_10_**	**434.3934**		**435**	**261; 243**	**243; 165**	**215; 161**	Eucalyptus [72]
**59**	**Flavone**	**Calycosin-7-** *O* **-beta-D-glucoside**	**C_22_H_22_O_10_**	**446.4041**		**447**	**285**	**270; 225; 145**	**242; 152**	*Astragali radix* [99]; [100]; Huolisu Oral Liquid [57];
**51**	**Flavone**	**Acacetin** *O* **-glucoside**	**C_22_H_22_O_10_**	**446.4041**		**447**	**285**	**269; 227; 145**	**241**	Mexican lupine species [58]
**52**	**Flavonol**	**Kaempferol-3-** *O* **-hexoside**	**C_21_H_20_O_11_**	**448.3769**		**449**	**329; 203**	**303; 257; 203; 185; 157**		Andean blueberry [43]; vinery products [52]; *F. glaucescens* [53]; *Rhus coriaria* [64]; *Punica granatum* [77]; *Cytisus multiflorus*; *Malva sylvestris* [96]
**53**	**Anthocyanin**	**Cyanidin-3-***O***-glucoside** [Cyanidin 3-*O*-beta-D-Glucoside; Kuromarin]	**C_21_H_21_O_11_+**	**449.3848**		**449**	**287**	**213; 175**	**213; 185; 141**	Triticum [40,101]; *acerola* [60]; Rice [65]; *Clidemia rubra* [82]; Rapeseed petals [71]; *Vigna sinensis* [102]; *Vitis labrusca* [103]
**54**	**Anabolic steroid**	**Vebonol**	**C_30_H_44_O_3_**	**452.6686**		**453**	**444; 389; 340; 276**	**435; 395; 336; 259**	**417; 331; 268**	*Rhus coriaria* [64]; *Hylocereus polyrhizus* [104]
**55**	**Anthocyanin**	**Pelargonidin 3-O-(6-O-malonyl-beta-D-glucoside)**	**C_24_H_23_O_13_**	**519.4388**		**519**	**271**	**215; 153**	**197**	*Gentiana lutea* [105]; Wheat [101]; Strawberry [106]
**56**	**Indole sesquiterpene alkaloid**	**Sespendole**	**C_33_H_45_NO_4_**	**519.7147**		**520**	**184; 502**	**166**		*Rhus coriaria* [64]; *Hylocereus polyrhizus* [104]
**57**	**Flavonol**	**Kaempferol diacetyl hexoside**	**C_25_H_24_O_13_**	**532.4503**		**533**	**285**	**270; 229; 145**	**242; 224; 152**	*A. cordifolia* [53]
**58**	**Flavone**	**Acacetin** *O* **-glucoside malonylated**	**C_25_H_24_O_13_**	**532.4503**		**533**	**285**	**269; 228; 145**	**196; 152**	Mexican lupine species [58]
**59**	**Condensed tannin**	**Procyanidin A-type dimer**	**C_30_H_24_O_12_**	**576.501**		**577**	**547; 493; 425; 245; 181**	**217**	**189; 161**	*Vaccinium macrocarpon* [73]; grape juice [107]; pear [108]
**60**	**Condensed tannin**	**Proanthocyanidin B1** [Procyanidin B1; Procyanidin Dimer B1; (−)-epicatechin-(4beta->8)-(+)-catechin; Epicatechin-(4beta->8)-ent-epicatechin]	**C_30_H_26_O_12_**	**578.5202**		**579**	**409; 343; 291; 247; 205**	**287; 259; 203; 163**	**245**	*Camellia kucha* [37]; millet grains [41]; *Vigna inguiculata* [49]; vinery products [52]; Andean blueberry [43]; *Vaccinium macrocarpon* [73]; strawberry [74]; grape juice [107]; pear [108]; *Senna singueana* [109]
**61**	**Condensed tannin**	**Procyanidin B2** [Epicatechin-(4beta->8)-epicatechin]	**C_30_H_26_O_12_**	**578.5202**		**579**	**427; 291; 247; 211**	**408; 327; 227; 139**	**379; 287; 257; 163**	millet grains [41]; *F. esculentum* [110]; Red wines [51]; blackberry [111]
**62**	**Steroidal alkaloid**	**Alpha-chaconine**	**C_45_H_73_NO_14_**	**852.0594**		**852**	**706**	**560**	**398**	Potato [39,112,113,114]
**63**	**Steroidal alkaloid**	**Solanidadiene solatriose**	**C_45_H_73_NO_15_**	**868.9588**		**868**	**706; 661; 560; 477**	**560; 398**	**382; 327**	Potato [113]

**Table 2 molecules-27-08228-t002:** Polyphenolic compounds identified in seven varieties of soybean.

№	Class of Compound	Identified Compound	Formula	k-569 (China)	k-5586 (Sweden)	k-5367 (China)	k-5373 (China)	k-11538 (Russia)	k-11559 (Russia)	Primorskaya-86 (Russia)
**1**	**Isoflavone**	**Daidzein** [4′,7 -Dihydroxyisoflavone; Daidzeol]	**C_15_H_10_O_4_**							
**2**	**7-hydroxyisoflavone**	**Formononetin** [Biochanin B; Formononetol]	**C_16_H_12_O_4_**							
**3**	**Flavone**	**Apigenin**	**C_15_H_10_O_5_**							
**4**	**7-hydroxyisoflavone**	**Formononetin** [Biochanin B; Formononetol]	**C_16_H_12_O_4_**							
**5**	**Flavone**	**Apigenin**	**C_15_H_10_O_5_**							
**6**	**Flavone**	**Acacetin** [Linarigenin; Buddleoflavonol]	**C_16_H_12_O_5_**							
**7**	**Flavone**	**6,7-Dihydroxy-4**′-**methoxyisoflavone**	**C_16_H_12_O_5_**							
**8**	**Flavone**	**Chrysoeriol** [Chryseriol]	**C_16_H_12_O_6_**							
**9**	**Flavone**	**5,7-Dimethoxyluteolin**	**C_17_H_14_O_6_**							
**10**	**Trihydroxyflavone**	**Jaceosidin**	**C_17_H_14_O_7_**							
**11**	**Trihydroxyflavone**	**5,7-Dimethoxy-3,3′,4**′-**trihydroxyflavone**	**C_17_H_14_O_7_**							
**12**	**Flavone**	**Syringetin**	**C_17_H_14_O_8_**							
**13**	**Flavone**	**5,6-Dihydroxy-7,8,3′,4**′-**tetramethoxyflavone**	**C_19_H_18_O_8_**							
**14**	**Flavone**	**Apigenin-7-*O*-glucoside**	**C_21_H_20_O_10_**							
**15**	**Flavone**	**Calycosin-7-*O*-beta-D-glucoside**	**C_22_H_22_O_10_**							
**16**	**Flavone**	**Acacetin *O*-glucoside**	**C_22_H_22_O_10_**							
**17**	**Flavone**	**Acacetin *O*-glucoside malonylated**	**C_25_H_24_O_13_**							
**18**	**Flavonol**	**Kaempferol**	**C_15_H_10_O_6_**							
**19**	**Flavonol**	**Quercetin**	**C_15_H_10_O_7_**							
**20**	**Flavonol**	**Rhamnetin I**	**C_16_H_12_O_7_**							
**21**	**Flavonol**	**Isorhamnetin**	**C_16_H_12_O_7_**							
**22**	**Flavonol**	**Myricetin**	**C_15_H_10_O_8_**							
**23**	**Flavonol**	**Myricetin 5-Methyl ether** [5-O-Methylmyricetin]	**C_16_H_12_O_8_**							
**24**	**Flavonol**	**Dihydrokaempferol-3-*O*-rhamnoside**	**C_21_H_22_O_10_**							
**25**	**Dihydroflavonol**	**Aromadendrin 7-O-rhamnoside**	**C_21_H_22_O_10_**							
**26**	**Flavonol**	**Kaempferol-3-*O*-hexoside**	**C_21_H_20_O_11_**							
**27**	**Flavonol**	**Kaempferol diacetyl hexoside**	**C_25_H_24_O_13_**							
**28**	**Flavan-3-ol**	**Epiafzelechin** [(epi)Afzelechin]	**C_15_H_14_O_5_**							
**29**	**Flavan-3-ol**	**Catechin**	**C_15_H_14_O_6_**							
**30**	**Flavan-3-ol**	**(epi)catechin**	**C_15_H_14_O_6_**							
**31**	**Flavanone**	**Hesperitin** [Hesperetin]	**C_16_H_14_O_6_**							
**32**	**Anthocyanin**	**Pelargonidin [Pelargonidol chloride]**	**C_15_H_11_O_5+_**							
**33**	**Anthocyanin**	**Cyanidin-3-*O*-glucoside**	**C_21_H_21_O_11_+**							
**34**	**Anthocyanin**	**Pelargonidin 3-O-(6-O-malonyl-beta-D-glucoside)**	**C_24_H_23_O_13_**							
**35**	**Condensed tannin**	**Procyanidin A-type dimer**	**C_30_H_24_O_12_**							
**36**	**Condensed tannin**	**Proanthocyanidin B1**	**C_30_H_26_O_12_**							
**37**	**Condensed tannin**	**Proanthocyanidin B2**	**C_30_H_26_O_12_**							
**38**	**Phenolic acid**	**Ferulic acid**	**C_10_H_10_O_4_**							
**39**	**Phenolic acid**	**Ellagic acid**	**C_14_H_6_O_8_**							
**40**	**Phenolic acid**	**Caffeic acid derivative**	**C_16_H_18_O_9_Na**							
**41**	**Phenolic acid**	**Salvianolic acid D**	**C_20_H_18_O_10_**							
**42**	**Phenolic acid**	**Ellagic acid pentoside**	**C_19_H_14_O_12_**							
**43**	**Stilbene**	**Resveratrol**	**C_14_H_12_O_3_**							
**44**	**Hydroxycoumarin**	**Umbelliferone hexoside**	**C_15_H_16_O_8_**							
**45**	**Lignan**	**Matairesinol**	**C_20_H_22_O_6_**							

## Data Availability

Not applicable.

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
