# Peer review of "Autofluorescence-Based Investigation of Spatial Distribution of Phenolic Compounds in Soybeans Using Confocal Laser Microscopy and a High-Resolution Mass Spectrometric Approach"

_molecules, 2022, doi:10.3390/molecules27238228_

Round 1

Reviewer 1 Report

The manuscript from Razgonova et al reports the results of auto fluorescence microscopy and LC/MS analysis of seven different strains of soybeans. The microscopy results show localisation of some classes of compounds within the seeds. The LC-MS analysis identified a large number of compounds and the occurrence of these in the different varieties was noted.

The manuscript does not propose or test any hypothesis. It simply reports a collection of results. More in depth analysis of the results would make this a stronger paper. For example, does the presence of different compounds in different stains make one more nutritionally beneficial than the others? Does the localisation of the compounds affect the availability of these compounds when consumed or processed? Were the amounts of the compounds present quantified?

There is little wrong with the manuscript but it shows little intellectual analysis.

Author Response

Dear reviewer!

Thank you so much for your interest.

The manuscript does not propose or test any hypothesis. It simply reports a collection of results. More in depth analysis of the results would make this a stronger paper. For example, does the presence of different compounds in different stains make one more nutritionally beneficial than the others? Does the localization of the compounds affect the availability of these compounds when consumed or processed? Were the amounts of the compounds present quantified?

There is little wrong with the manuscript but it shows little intellectual analysis.

All sections of the manuscript have been supplemented and better structured. We rewrote the text and put some emphasis on the importance of phenolic compounds for the development of resistance, UV resistance, resistance to viral and fungal diseases. These are important breeding traits that help improve the quality of varieties and it is not always correct to focus on the breeding process, only on the visual color of seeds or tissues. We have shown that the use of confocal laser microscopy with the use of autofluorescence makes it possible to localize different classes of compounds and the level of their accumulation. This is a quick and inexpensive method that allows breeders to significantly improve the quality of seed selection when breeding new varieties.

Reviewer 2 Report

This paper reports a comparative analysis of bioactive compounds in different varieties of soybean from different countries by using both confocal laser microscopy and mass spectrometry. The work is interesting but suffers of serious flaws both from a substantial and formal point of view: the paper is not well organized and the results are confused.

The introduction is poor, the state of the art is missing, the novelty is not highlighted, the methodology is not well described; on the contrary the conclusion paragraph is too long and does not clarify the spirit of the paper.

In general, the article has potential, however the form needs to be improved, both linguistically and in results. The "results and discussions" section must be improved and clarified based on the results obtained and a more complete description of the position of polyphenols and other compounds in soybeans must be provided. In addition, to the clarification of the question referred to in line 137-138 “and the likelihood of their involvement in resistance and yield” which is not reported in the conclusions. In the “introduction” section there is no reference on phenolic compounds, besides a brief mention, and some information on classes and structures should be provided.

The part relating to mass spectrometry must be improved with more detailed information on the acquisition mode used for identification (MRM?), also the ionization mode used for the various compounds must be considered; in fact, since initially it is claimed to work on “negative ionization”, while "table 1" also refers to “positive ionization” for several analytes, such as Proanthocyanidin B1. In addition, "table 2" shows information relating only to the arrangement of phenolic compounds in the seed, while the previous analyzes also show the presence of other kinds of compounds not reported in the “conclusions” section (Line 281-296). In my opinion, the first part of the "conclusions" (Line: 281-305) should be reported in the “introduction” section and the conclusions should provide clear and concise information on the potential of the proposed study.

Line 92-94: not very clear. How much solvent was used?

Row 107-112: the extraction part should be inserted after maceration (2.3. Fractional maceration).

Line 108: "4 C" in "4 °C"

Line 119. "the temperature was 50°C and the total flow rate was 0.25 ml min-1" in "the temperature was 50 °C and the total flow rate was 0.25 ml min-1"

Line 155: "various phenolic and polyphenolic compounds" in "various phenolic compounds"

Row 156-158: Phenolic acids

Line 230: “HPLS-MS/MS” in “HPLC-MS/MS”

Reference list: 106 references seem too much for a paper; furthermore the introduction is very poor regarding the state of the art. Please reduce the number of citation and rationalize the introduction

Author Response

Dear reviewer!

Thank your so much for brilliant review! 

  1. The introduction is poor, the state of the art is missing, the novelty is not highlighted, the methodology is not well described; on the contrary the conclusion paragraph is too long and does not clarify the spirit of the paper. In general, the article has potential, however the form needs to be improved, both linguistically and in results. The "results and discussions" section must be improved and clarified based on the results obtained and a more complete description of the position of polyphenols and other compounds in soybeans must be provided.

The introduction and description of the results are significantly expanded. The conclusion is shortened and contains only specific conclusions from the work.

  1. In addition, to the clarification of the question referred to in line 137-138 “and the likelihood of their involvement in resistance and yield” which is not reported in the conclusions.

This unfortunate phrase has been deleted.

  1. In the “introduction” section there is no reference on phenolic compounds, besides a brief mention, and some information on classes and structures should be provided.

Information about the classes of phenols in soybeans and their roles has been added to the introduction.

  1. The part relating to mass spectrometry must be improved with more detailed information on the acquisition mode used for identification (MRM?), also the ionization mode used for the various compounds must be considered; in fact, since initially it is claimed to work on “negative ionization”, while "table 1" also refers to “positive ionization” for several analytes, such as Proanthocyanidin B1. In addition, "table 2" shows information relating only to the arrangement of phenolic compounds in the seed, while the previous analyzes also show the presence of other kinds of compounds not reported in the “conclusions” section (Line 281-296).

The method of mass spectrometric analysis was applied to confirm the conclusions made using fluorescence microscopy, and to be able to analyze which groups of compounds can be seen in different fluorescence ranges. Accordingly, in Table 2 there is already an analysis of compounds that can be detected by these two methods. We did this in order to show the strengths and possibilities in carrying out breeding work, even for such an inexpensive method as fluorescence microscopy. Since not every breeding station has the ability to analyze a large volume of samples using mass spectrometry methods. Since the main idea is to use the method of fluorescence microscopy in the selection of variety samples in breeding, the data of a complete biochemical analysis were not transferred to the conclusion.

Thanks also for the remark, indeed the analysis was carried out both in the mode of negative ions and in the mode of positively charged ions. We have corrected this part of the method description.

  1. In my opinion, the first part of the "conclusions" (Line: 281-305) should be reported in the “introduction” section and the conclusions should provide clear and concise information on the potential of the proposed study.

This paragraph has been moved to the introduction.

  1. Line 92-94: not very clear. How much solvent was used?

I don't understand what he doesn't understand. We have: "A solid-solvent ratio was 1:20" (now lines 121-122). Since we take 1 gram of soybean seeds as indicated, and the seeds are not strictly the same, the mkssa was, 1.05; 1.11; 1.07, etc., respectively, and the solvent was 1:20 (21 ml, 22.2 ml, 21.4 ml). It is standard practice to write ratios to display exact proportions.

  1. Row 107-112: the extraction part should be inserted after maceration (2.3. Fractional maceration).

Transferred to maceration (Line 123-127).

  1. Line 108: "4 C" in "4 °C"

Fixed (Line 124).

  1. Line 119. "the temperature was 50°C and the total flow rate was 0.25 ml min-1" in "the temperature was 50 °C and the total flow rate was 0.25 ml min-1"

Fixed (Line 148).

  1. Line 155: "various phenolic and polyphenolic compounds" in "various phenolic compounds"

This sentence is deleted.

  1. Row 156-158: Phenolic acids

This sentence is deleted.

  1. Line 230: “HPLS-MS/MS” in “HPLC-MS/MS”

Fixed. (Line 247).

Reviewer 3 Report

molecules-1934383

The manuscript entitled "Autofluorescence-based investigation of spatial distribution of 2 some groups of phytochemical substances in the soybeans us-3 ing confocal laser microscopy and a high-resolution mass spec-4 trometric approach for the comprehensive characterization of 5 polyphenols”

Some comments and suggestions are made, as follows:

1.       The manuscript is not well structured and presented in a clear way and supported by extensive and validated data.

2.       The title of this paper is lengthy, it needs to be revised.

3.       List of keywords needs to be updated. Avoid the redundant terms which are already coated in the title and abstract

4.       The authors should highlight the novelty of the work in Introduction, so that it is clearly visible to the reader.

5.       More detailed discussion on the optical microscopy and mass spectrometric results should be accordingly added.

6.       The conclusion should paraphrase to be more comprehensive and summarized. For example, what are authors own viewpoints? What are the major findings and how they are addressing the left behind research gaps and current challenges?

7.       There are some grammatical errors in the manuscript. The authors need to carefully correct the errors in the revised manuscript.

I recommend the paper to be considered for publication in the Molecules after a major revision.

Author Response

Dear reviewer!

Thank you so much for the brilliant review.

  1. The manuscript is not well structured and presented in a clear way and supported by extensive and validated data.

 Improved.

  1. The title of this paper is lengthy, it needs to be revised.

We suggest shortening the title as it now appears in the article: Autofluorescence-based investigation of spatial distribution of phenolic compounds in the soybeans using confocal laser microscopy and a high-resolution mass spectrometric approach.

  1. List of keywords needs to be updated. Avoid the redundant terms which are already coated in the title and abstract.

Improved.

4. The authors should highlight the novelty of the work in Introduction, so that it is clearly visible to the reader.

The introduction is added.

  1. More detailed discussion on the optical microscopy and mass spectrometric results should be accordingly added.

The microscopy results are completely rewritten.

  1. The conclusion should paraphrase to be more comprehensive and summarized. For example, what are authors own viewpoints? What are the major findings and how they are addressing the left behind research gaps and current challenges?

The conclusion is abbreviated and contains only specific conclusions from the work and prospects for their use.

  1. There are some grammatical errors in the manuscript. The authors need to carefully correct the errors in the revised manuscript.

Checked.
